# Unlocking Teacher Professional Performance: Exploring Teaching Creativity in Transmitting Digital Literacy, Grit, and Instructional Quality

**Jafriansen Damanik** *[ID] **and Widodo Widodo**

Social Science Education Department, Postgraduate Faculty, Universitas Indraprasta PGRI, Jakarta 12530, Indonesia; widodo@unindra.ac.id
* Correspondence: jafriansen.damanik@unindra.ac.id

**Abstract:** Schools need teachers' professional performance to ensure the quality of educational output. Therefore, this research explores teachers' professional performance based on digital literacy, grit, and instructional quality mediated by teaching creativity. The research participants are 465 junior- and high-school teachers in Indonesia. Structural equation modeling (SEM) is utilized in the data analysis, along with common method bias and correlational and descriptive analyses. The results show a significant relationship between digital literacy, grit, and instructional quality and teaching creativity and teacher professional performance. Teaching creativity also has a significant relationship with teachers' professional performance and mediates the influence of digital literacy, grit, and instructional quality on teachers' professional performance. This finding promotes a new empirical model of the causal relationship between digital literacy, grit, instructional quality, and teacher professional performance through teaching creativity. Consequently, it is proposed that teaching creativity, grit, digital literacy, and high-quality instruction can all improve teachers' professional performance. Therefore, in order to advance teachers' professional performance in the future, practitioners and researchers should discuss, modify, and possibly even adopt the new empirical model.

**Keywords:** digital literacy; grit; instructional quality; teaching creativity; professional performance

## 1. Introduction

Teachers' professional performance is crucial for students and schools. Professional performance is a series of activities carried out by employees to complete tasks or work according to their position and responsibilities efficiently, effectively, politely, empathetically, and lovingly to achieve organizational goals. Recent studies have indicated that teachers' professional performance significantly affects students' academic achievement [1,2]. Another study demonstrated that it impacts school effectiveness [3]. These findings confirm that teachers' professional performance is a crucial factor determining the quality of schools, including the graduates produced. Therefore, the high academic achievements of students also reflect the high professional performance of teachers. On the other hand, low academic achievement can indicate suboptimal professional performance of teachers. Thus, when the results of the Program for International Student Assessment (PISA) evaluation in 2022 showed a decrease in Indonesia's scores in terms of reading, mathematics, and science abilities, it could also be read as an illustration of the decline in teacher professional performance, which was influenced by, among other things, the COVID-19 pandemic. This phenomenon must urgently be investigated, primarily through the perspectives of digital literacy, grit, instructional quality, and teaching creativity. The results of recent research show that teachers' professional performance is influenced by digital literacy [4,5], grit [6,7], instructional quality [8], and teaching creativity [9,10]. Other studies show that teaching creativity, as well as influencing teachers' professional performance, is also influenced by digital literacy [11,12], grit [13,14], and instructional quality [15]. These empirical facts show

the opportunity for teaching creativity as a mediator of the causal relationship between digital literacy, grit, and instructional quality and professional performance. In this way, a research theoretical/conceptual framework of the influence of digital literacy, grit, and instructional quality on professional performance can be built through the mediation of teaching creativity.

However, several other studies show contradictory results. For example, Fabelico and Afalla [16] revealed that grit has no significant effect on performance. By contrast, Nussbaum et al. [17] found empirical evidence showing that creativity affects grit. Lee and Lee [18] found that teaching creativity is an essential determinant of creative class management and an indicator of instructional quality. Other studies also prove that professionalism influences creativity [19,20]. Lastly, Hassan et al. [21] established that creative teaching greatly influences the quality of learning, along with parental guidance, learning strategies, and student interest. The inconsistencies in the literature produce research gaps that can disrupt the theoretical framework, which requires scientific clarification to avoid confusion among academics/researchers and practitioners. Therefore, this research addresses the problem by exploring teachers' professional performance based on digital literacy, grit, and instructional quality mediated by teaching creativity.

## 2. Literature Review

### 2.1. Professional Performance

Conceptually, professional performance is a set of duties or behaviors performed with decency, responsibility, empathy, and compassion in order to fulfill job requirements or accomplish organizational goals [22]. It also refers to how an employee carries out the activities related to their position [23]. Professional performance can be assessed with various methods and techniques. It is essential for organizations to accurately evaluate the performance of their employees in order to make strategic decisions and improve their overall effectiveness [24]. Professional performance assessment measures how employees efficiently carry out their duties and responsibilities [25]. It also includes evaluating individuals' or groups' potential, skills, behavior, and overall performance [26]. The goal of professional performance assessment is to determine the level of achievement and make decisions for continuous improvement. In the context of schools, especially teaching, professional performance can be measured with three indicators: subjects, didactics, and pedagogy. Subjects are related to teachers' mastery, study, evaluation, and updating of knowledge regarding the details of the subject matter that they teach. Didactics refers to transmitting knowledge to students using various teaching methods, considering classroom conditions and student characteristics. Lastly, pedagogy refers to orientation and interest in students' actual problems, interest in solving educational problems inside and outside the classroom, as well as respectful, ethical, and consistent actions [27].

### 2.2. Digital Literacy and Professional Performance

An individual's ability to use digital technology to find, evaluate, produce, and communicate information is known as digital literacy [28]. It is the capacity to comprehend digital and computer resources [29]. According to certain academics, digital literacy and cognitive ability are closely related [30,31]. Mars [32] states that there are three phases of digital literacy: digital usage (using acquired skills in an applied context), digital competence (gaining a variety of skills), and digital transformation (using acquired skills to generate creativity and innovation). Digital literacy plays a vital role in increasing the effectiveness of the educational process and preparing students for success in modern society [33]. Higgins et al. [34] claim that a lack of digital literacy can lead to a lack of self-control and abnormal behavior online. Children who lack digital literacy may also develop a device addiction [35,36]. According to a study by Mohammadyari and Singh [37], a person's degree of digital literacy can affect how well their students perform by enabling e-learning and lessening the detrimental effects of online activities. Therefore, digital literacy is not only important for students but also vital for teachers, who are the main

actors in providing education. Hence, teachers need to have a high level of digital literacy in order to be able to engage in professional activities centered on digital technology and improve educational standards. It includes skills and competencies that enable them to integrate existing and new technologies into the teaching and learning process. Research results show that teacher digital competence is relatively varied [38], so it needs to be improved continuously.

For this reason, researchers, policymakers, and stakeholders in education should prioritize the teaching of digital literacy [39]. By incorporating digital literacy into curricula, the government and legislators of today have turned it into a valuable tool for the educational system. It enables educational institutions to prepare graduates and future workers for supportive and technology-based workplaces. New digital technology is increasingly being discovered and used, so digital literacy is also developing. Digital literacy includes reading and writing, with technology being the main characteristic of communication patterns [40,41]. Therefore, when teachers have an adequate (high) level of digital literacy, including digital competence, usage, and transformation, it can drive their professional performance. Several prior studies also indicated that digital literacy is related to teachers' professional performance [4,5,42–46]. Therefore, we can propose the first hypothesis (H):

**H1:** *Digital literacy directly relates to teachers' professional performance.*

*2.3. Grit and Professional Performance*

Grit combines passion and perseverance for long-term goals [47]. It is a beneficial noncognitive skill linked to enhanced mental health and favorable social outcomes [48]. Grit is the result of tenacity and passion for long-term objectives [49]. It also refers to a psychological factor linked to the long-term achievement of high-level goals that emphasizes perseverance as a sign of success [50,51]. Widodo et al. [52] define grit as the propensity to work hard, persistently, and resiliently for an extended period of time in the face of obstacles, setbacks, and failures in order to accomplish significant goals that have personal significance. Grit is measured by two things: tenacity of effort and consistency of interest [53]. Grit is an empirical predictor and result of success in school, in the workplace, and in one's personal life [6]. Within the educational setting, it affects students' self-reported grades [49,54], academic engagement, successful educational outcomes [42], and academic achievements [55–58]. Several previous research results show the effectiveness of grit in influencing various aspects of individual, group, and organizational life. Unfortunately, people have recently begun to question the existence of grit. Grit is suspected of being racist in higher education circles in the United States [59]. However, this issue is still very limited to specific regions (for example, the United States), so its resonance does not interfere with the development of grit studies in various parts of the world, including this study. Empirical evidence suggests that grit impacts professional performance. Scholars claim that grit influences professional performance [6,7,22,60–64]. Hence, the second hypothesis is proposed:

**H2:** *Grit directly relates to teachers' professional performance.*

*2.4. Instructional Quality and Professional Performance*

Instructional quality refers to the evaluation and improvement of teaching practices in order to enhance student learning outcomes. It is a multidimensional construct that encompasses various aspects of teaching. The criteria for instructional quality may vary depending on the context and the stakeholders involved. Evaluations of instructional quality can be conducted by different groups, including advisors and pre-service teachers, who may have different perspectives and priorities. The concept of instructional quality is also relevant in online education, such as massive open online courses, where the design and implementation of instructional strategies play a crucial role in the effectiveness of the courses. Overall, instructional quality is a complex and multifaceted concept that requires

careful consideration and assessment to enhance learning and teaching experiences [65,66]. Instructional quality also reflects classroom teaching characteristics that teachers manage and observe in a way that makes sense and is supported by empirical evidence that aligns with the development of student learning outcomes [67]. To date, finding an adequate definition of instructional quality has been quite difficult. As a guide, in this study, instructional quality refers to the teacher's behavior in delivering lesson material in class by utilizing various learning methods and media.

Instructional quality has various indicators: classroom management, student support, and cognitive activation [68–71]. Using allotted class time wisely in order to maintain order and uphold rules is known as classroom management [69]. Student support encompasses the assessment of students' learning, providing opportunities for customization and differentiation, and establishing a positive learning environment [70]. In terms of teaching strategies and chosen learning tasks, cognitive activation describes whether or not students are forced to engage in higher-order thinking [68,71]. Teacher instructional quality has been acknowledged as an essential source of variation in student learning and achievement [72,73]. Teacher quality, including competence and performance in the classroom, has a significant impact on student learning outcomes [8]. Additionally, teaching quality relates to performance [74]. Accordingly, we can formulate the third hypothesis:

**H3:** *Instructional quality directly relates to teachers' professional performance.*

### 2.5. Teaching Creativity and Professional Performance

Teaching creativity describes the capacity of educators to reframe novel or creative concepts pertaining to approaches, strategies, tactics, forms, and resources for instructional activities during the learning process [13]. It illustrates how well educators are able to support and develop their students' creative thinking during the educational process. It entails teaching pupils specific subject matter, logical analysis, and support for their imaginative and affective reactions. Effective teaching in all subject areas requires creative teaching, which calls on educators to constantly reflect on and integrate the many aspects of their work-related learning. It also involves providing rich experiences and supporting students by using innovative approaches to enhance their creative skills. Teachers who use pedagogical strategies that foster task motivation, domain-specific skills, and creativity-relevant processes can help students develop their creative abilities. Therefore, it is recommended that teacher training and development explicitly focus on fostering creative teaching approaches [75,76].

Teaching creativity has several roles and benefits. It allows teachers to respond to learners' diverse interests and needs while adapting to the changing demands of education systems [77]. Creative teaching approaches can be fostered through various learning experiences, leading to increased understanding and use of creative teaching methods among teachers [78]. Creative teaching can become a part of a teacher's identity and positively impact their first year of teaching [76]. Fostering creativity in the classroom is valuable as it encourages thinking outside the box, overcoming societal pressures, and promoting involvement and collaboration. Teaching creativity is essential for teachers when providing their students with learning material because it impacts their learning outcomes. Moreover, students benefit greatly when teachers have high levels of creativity in their teaching. Creative teaching approaches can enhance student learning and understanding [75,79] because they can express and realize the potential of their thinking power, resulting in innovative and exciting teaching methods [76]. Creative teaching can increase students' learning motivation and interest in the subject matter being taught [79]. It also nurtures a creative disposition and improves creative thinking skills [80]. Under such conditions, teaching creativity is an asset that allows teachers to meet high professional performance standards. Therefore, teachers with adequate fluency, flexibility, originality, elaboration, and redefinition in teaching [13,81] tend to meet professional performance standards easily.

Researchers also claim that teaching creativity has a significant relationship with professional performance [9,10,82–84]. Therefore, the fourth hypothesis can be proposed:

**H4:** *Teaching creativity directly relates to teachers' professional performance.*

### 2.6. Digital Literacy and Teaching Creativity

Teaching creativity not only influences teachers' professional performance but is also influenced by digital literacy. Several recent studies have convincingly proven that digital literacy has a positive relationship with teaching creativity [11,12,85–89]. In order to teach creativity, digital literacy is essential because it improves teachers' abilities and allows them to stay up-to-date with the rapidly evolving technological landscape. Technology use in the classroom fosters the development of many abilities, including productivity, imagination in collaboration, problem-solving, curiosity, and other traits that boost creativity. Additionally, it improves educators' motivation, collaboration, and creativity [90]. Using technology in the classroom fosters a variety of abilities, including increased creativity, productivity, curiosity, teamwork, and inventiveness [91]. Furthermore, knowledge and information are viewed as giving businesses and service providers a competitive edge in the workplace [92]. Workers with information literacy skills—the ability to find, assess, and use information—are becoming a more crucial strategic asset for businesses [93]. Moreover, digital pedagogies that leverage embodied liveness, playful interactivity, and generative curiosity can be harnessed in digital learning to support students in taking risks, collaborating, and working responsively in diverse situations [94]. Teachers use visual and digital texts as a basis for composition, helping children find their own voices and develop autonomy and agency [12]. Digital literacy empowers educators and students alike to explore new possibilities, collaborate, and create in the digital age. Based on previous studies and the argument above, we can formulate the fifth hypothesis:

**H5:** *Digital literacy directly relates to teaching creativity.*

### 2.7. Grit and Teaching Creativity

Teaching creativity is also influenced by grit. Results in the literature indicate that grit has a significant relationship with teaching creativity [13,14,95]. Sun [96] also states that grit is essential for teaching creativity. Gritty educators are more inspired to handle difficult situations [97]. Grit also significantly predicts creativity [98]. In this case, grit refers to stamina and perseverance in facing challenges [99]. It indicates that teachers with high consistency of interests and persistence of effort over a long period would have the additional energy to find alternative ways of teaching, which are expressed in fluency, flexibility, originality, elaboration, or redefinition. Therefore, in this context, grit is an important determinant of teaching creativity. According to prior studies and the argument above, we can propose the sixth hypothesis:

**H6:** *Grit directly relates to teaching creativity.*

### 2.8. Instructional Quality and Teaching Creativity

Teaching creativity is also influenced by the quality of the teaching. Investigations conducted by Lowel [15] prove that instructional quality influences teaching creativity. Learning strategies that support creative and innovative education, such as design-based learning, problem-solving, and project-based learning, can also increase creativity in teaching [100]. This shows that instructional quality that supports teaching creativity is essential for developing creativity and the quality of education [100,101]. In line with this, instructional management models have been shown to affect the development of students' creative thinking positively and significantly [102]. Moreover, instructional quality impacts teaching creativity [103]. Therefore, it is crucial to research the relationship between them. Accordingly, we can propose the seventh hypothesis:

**H7:** *Instructional quality directly relates to teaching creativity.*

## 3. Materials and Methods

### 3.1. Participants

The study sample across three provinces—Jakarta, West Java, and Banten—consisted of 465 Indonesian junior- and high-school teachers. Women (67.96%) make up the majority of the age group, with 35.91% being between 26 and 35 years old, 77.20% holding bachelor's degrees, and 35.27% having at least 16 years of work experience.

### 3.2. Procedure and Materials

This study employs a survey method in conjunction with a quantitative approach. A questionnaire with five options—strongly disagree/never (score = 1), disagree/rarely (score = 2), neutral/sometimes (score = 3), agree/often (score = 4), and strongly agree/always (score = 5)—was used to gather data using a Likert scale. Google Forms was used for the online survey, and WhatsApp was used to share the results. Researchers created the questionnaire based on the theoretical dimensions or indicators of experts in the literature. The digital literacy indicators were digital competence (DC), digital usage (DU), and digital transformation (DT) [32]; for grit, they were consistency of interests (COI) and persistence of efforts (POE) [53]; for instructional quality, they were classroom management (CM), student support (SS), and cognitive activation (CA) [68–71]; for teaching creativity, they were fluency (Flue), flexibility (Flex), originality (Orig), elaboration (Elab), and redefinition (Rede) [13,81]; and for professional performance, they were subjects (Subj), didactics (Dida), and pedagogy (Peda) [27]. As presented in Appendix A, digital literacy consists of nine items with a corrected item–total correlation coefficient (CITCC) range of 0.441–0.794 and an alpha coefficient (AC) of 0.897. Grit consists of eight items with a CITCC range of 0.469–0.873 and an AC of 0.883. Instructional quality consists of twelve items with a CITCC range of 0.416–0.819 and an AC of 0.901. Teaching creativity consists of ten items with a CITCC range of 0.452–0.763 and an AC of 0.859. Lastly, professional performance comprises nine items with a CITCC range of 0.555–0.903 and an AC of 0.912. All items have a CITCC > 0.361, and all variables have an AC > 0.70; therefore, it is valid and reliable as a research instrument [104,105].

Additionally, this study carried out a common method bias (CMB) test to ensure that the research data were not contaminated with data bias. A concern in preliminary research is that cross-sectional surveys, particularly those using self-report questionnaires such as the one adopted in the present investigation, tend to overlook CMB, a potential source of measurement error. The difference between the genuine correlation among constructs (variables) and the observed relationship determined using common method variance (CMV) is measured with the CMB [52]. The disparity between perceived and actual correlations, facilitated by CMV, poses a significant threat to the validity and reliability of the research findings [106]. In response to this concern, Fuller et al. [107] recommended a combination of statistical and procedural improvements for controlling and minimizing CMV. The present research adopted two methods due to this issue. Procedural improvements include (1) formulating distinct statements and alternative answers or responses for predictor and criteria variables; (2) incorporating an introductory section in the questionnaire with clear explanations or instructions to make respondents feel comfortable, including sentences indicating, for example, that there is no right or wrong answer or response and the data collected would be safe, protected, and used only for research purposes; and (3) conducting a pilot test with 30 people as a preliminary step. Additionally, statistical mechanisms such as the correlation matrix method and the Harman single-factor test were used to ensure improvements. The coefficient between a construct (variable) is less than 0.90, and the total variance extracted by one factor is 42.353%, less than the suggested threshold of 50%. It shows an absence of CMV (CMB) in the acquired data [108,109].

### 3.3. Data Analysis

Structural equation modeling (SEM), along with correlational and descriptive statistics, was used to analyze the data. A Student's test ($t$-test) was used to determine the direct significance of the path coefficient correlation, and a Sobel (Z) test was used to determine the indirect relationship. SPSS version 22 was used for the common method biases (descriptive and correlational), and LISREL version 8.80 was used for the SEM analysis.

## 4. Results

### 4.1. Descriptive and Correlation Analysis

The mean value indicators of digital literacy are shown in the descriptive statistical analysis results for the five research variables, ranging from the lowest to the highest: DC, DU, and DT are 11.45, 11.81, and 12.26; for grit, COI and POE are 17.23 and 17.50; for instructional quality, CA, SS, and CM are 15.75, 17.19, and 18.47; for teaching creativity, Flue, Elab, Rede, Orig, and Flex are 8.33, 8.44, 8.47, 8.49, and 8.66; for professional performance, Peda, Dida, and Subj are 12.90, 12.94, and 13.18. The digital literacy indicators' standard deviation (SD) values are as follows, from lowest to highest: DC = 1.966, DU = 2.018, and DT = 2.136; grit: POE = 1.996 and COI = 2.287; instructional quality: CM = 1.755, SS = 2.005, and CA = 2.466; teaching creativity: Flex = 1.053, Orig = 1.132, Rede = 1.148, Elab = 1.175, and Flue = 1.234; and professional performance: Subj = 1.373, Dida = 1.409, and Peda = 1.460. Table 1 illustrates how the standard deviation values are typically lower than the mean values, indicating a good representation of the total data. At the $p < 0.01$ level, the correlation analysis result for all research variable indicators demonstrates a significant relationship with the indicators of other variables. It demonstrates how each indicator and every other indicator are related to each other.

**Table 1.** Descriptive and correlation statistics results.

| Indicators | Descriptive | | Correlation | | | | | | | | | | | | | | | |
|---|---|---|---|---|---|---|---|---|---|---|---|---|---|---|---|---|---|---|
| | Mean | SD | 1 | 2 | 3 | 4 | 5 | 6 | 7 | 8 | 9 | 10 | 11 | 12 | 13 | 14 | 15 | 16 |
| **Digital Literacy (X₁)** | | | | | | | | | | | | | | | | | | |
| 1. DC | 11.45 | 1.966 | 1.00 | | | | | | | | | | | | | | | |
| 2. DU | 11.81 | 2.018 | 0.67 ** | 1.00 | | | | | | | | | | | | | | |
| 3. DT | 12.26 | 2.136 | 0.44 ** | 0.65 ** | 1.00 | | | | | | | | | | | | | |
| **Grit (X₂)** | | | | | | | | | | | | | | | | | | |
| 4. COI | 17.23 | 2.287 | 0.29 ** | 0.20 ** | 0.15 ** | 1.00 | | | | | | | | | | | | |
| 5. POE | 17.50 | 1.996 | 0.39 ** | 0.36 ** | 0.27 ** | 0.62 ** | 1.00 | | | | | | | | | | | |
| **Instructional Quality (X₃)** | | | | | | | | | | | | | | | | | | |
| 6. CM | 18.47 | 1.755 | 0.14 ** | 0.19 ** | 0.20 ** | 0.27 ** | 0.21 ** | 1.00 | | | | | | | | | | |
| 7. SS | 17.19 | 2.005 | 0.29 ** | 0.32 ** | 0.26 ** | 0.16 ** | 0.17 ** | 0.39 ** | 1.00 | | | | | | | | | |
| 8. CA | 15.75 | 2.466 | 0.29 ** | 0.28 ** | 0.22 ** | 0.15 ** | 0.22 ** | 0.39 ** | 0.53 ** | 1.00 | | | | | | | | |
| **Teaching Creativity (Y₁)** | | | | | | | | | | | | | | | | | | |
| 9. Flue | 8.33 | 1.234 | 0.42 ** | 0.39 ** | 0.23 ** | 0.29 ** | 0.35 ** | 0.35 ** | 0.36 ** | 0.31 ** | 1.00 | | | | | | | |
| 10. Flex | 8.66 | 1.053 | 0.24 ** | 0.20 ** | 0.13 ** | 0.30 ** | 0.35 ** | 0.19 ** | 0.29 ** | 0.15 ** | 0.43 ** | 1.00 | | | | | | |
| 11. Orig | 8.49 | 1.132 | 0.36 ** | 0.31 ** | 0.19 ** | 0.23 ** | 0.35 ** | 0.23 ** | 0.32 ** | 0.23 ** | 0.58 ** | 0.54 ** | 1.00 | | | | | |
| 12. Elab | 8.44 | 1.175 | 0.36 ** | 0.38 ** | 0.24 ** | 0.22 ** | 0.36 ** | 0.24 ** | 0.35 ** | 0.28 ** | 0.48 ** | 0.45 ** | 0.66 ** | 1.00 | | | | |
| 13. Rede | 8.47 | 1.148 | 0.42 ** | 0.39 ** | 0.23 ** | 0.27 ** | 0.43 ** | 0.25 ** | 0.35 ** | 0.31 ** | 0.51 ** | 0.49 ** | 0.60 ** | 0.64 ** | 1.00 | | | |
| **Professional Performance (Y₂)** | | | | | | | | | | | | | | | | | | |
| 14. Subj | 13.18 | 1.373 | 0.42 ** | 0.39 ** | 0.38 ** | 0.35 ** | 0.43 ** | 0.34 ** | 0.32 ** | 0.37 ** | 0.48 ** | 0.35 ** | 0.42 ** | 0.45 ** | 0.53 ** | 1.00 | | |
| 15. Dida | 12.94 | 1.409 | 0.42 ** | 0.20 ** | 0.43 ** | 0.34 ** | 0.44 ** | 0.38 ** | 0.41 ** | 0.43 ** | 0.44 ** | 0.43 ** | 0.49 ** | 0.51 ** | 0.50 ** | 0.69 ** | 1.00 | |
| 16. Peda | 12.90 | 1.460 | 0.41 ** | 0.31 ** | 0.42 ** | 0.37 ** | 0.51 ** | 0.30 ** | 0.37 ** | 0.37 ** | 0.42 ** | 0.41 ** | 0.49 ** | 0.47 ** | 0.52 ** | 0.63 ** | 0.77 ** | 1.00 |

** $p < 0.01$.

### 4.2. Confirmatory Factor Analysis

Table 2 displays the measurement model estimate derived from confirmatory factor analysis. All indicators have factor loading values greater than 0.50, indicating validity. This implies that each indicator can measure or represent the corresponding construct (variables). Concurrently, the Cronbach's Alpha (CA), construct reliability (CR), and variance extracted (VE) values were used to assess the validity and reliability of the constructs. Good reliability and acceptable convergence are indicated by the CA and CR values of all constructs being greater than 0.70 and the VE values of all variables being greater than 0.50 [105].

**Table 2.** Results of the measurement model.

| Constructs | Indicators | Factor Loading | CA | CR | VE |
|---|---|---|---|---|---|
| Digital Literacy ($X_1$) | DC<br>DU<br>DT | 0.68<br>0.99<br>0.66 | 0.897 | 0.829 | 0.626 |
| Grit ($X_2$) | COI<br>POE | 0.88<br>0.71 | 0.883 | 0.725 | 0.526 |
| Instructional Quality ($X_3$) | CM<br>SS<br>CA | 0.54<br>0.74<br>0.73 | 0.901 | 0.713 | 0.518 |
| Teaching Creativity ($Y_1$) | Flue<br>Flex<br>Orig<br>Elab<br>Rede | 0.67<br>0.63<br>0.84<br>0.79<br>0.76 | 0.859 | 0.858 | 0.551 |
| Professional Performance ($Y_2$) | Subj<br>Dida<br>Peda | 0.75<br>0.92<br>0.85 | 0.912 | 0.880 | 0.711 |

### 4.3. Goodness of Fit

A high degree of suitability indicates a good fit, according to the goodness-of-fit (GOF) test, which assesses the theoretical model's correlation with empirical data. Nine out of the eleven criteria indicated a good fit, according to the results, while the other two had poor GOF. The criteria that were met were the RMSEA, GFI, NFI, NNFI, AGFI, CFI, RFI, normed chi-square, and PNFI; however, sig. probability and chi-square were not met. Hair et al. [105] noted that although there are difficulties with large samples—like the 465 individuals in this study—the GOF test results are still valid (fit) because nine out of the eleven samples satisfied the requirement.

### 4.4. Hypothesis Testing

The results from the hypothesis tests are presented in Figures 1 and 2 and summarized in Table 3. All hypotheses, from $H_1$ to $H_7$, were supported (significant) with a t value > t table at $\alpha$ = 0.01 (1.65). Specifically, digital literacy, grit, instructional quality, and teaching creativity are significantly related to professional performance, with a path coefficient ($\gamma / \beta$) = 0.22, 0.21, 0.28, and 0.35. Furthermore, digital literacy, grit, and instructional quality are also significantly related to teaching creativity, with a path coefficient ($\gamma$) = 0.20, 0.32, and 0.37. The strongest relationship between variables is instructional quality with teaching creativity, followed by teaching creativity with professional performance, and instructional quality with professional performance. The weakest relationship is digital literacy with teaching creativity, followed by grit and digital literacy with professional performance.

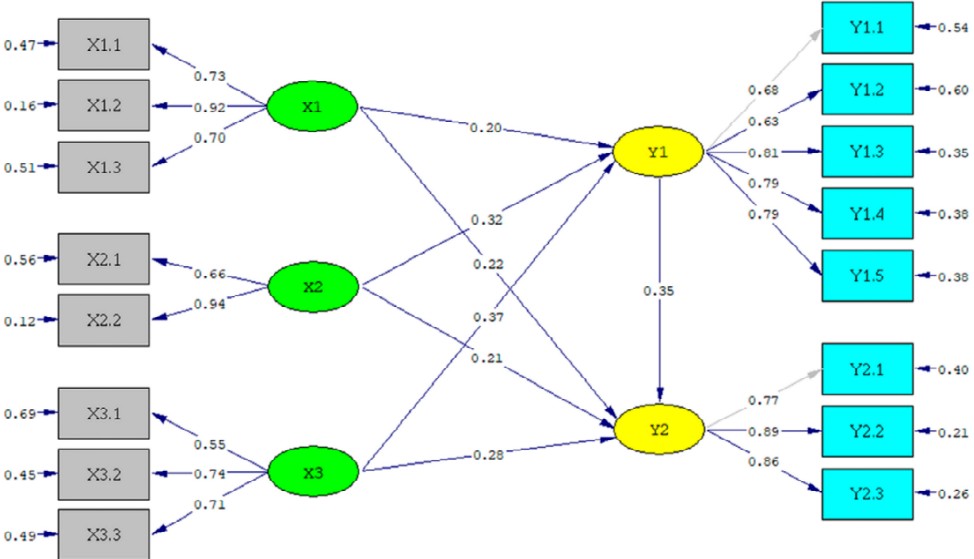

Chi-Square=288.04, df=94, P-value=0.00000, RMSEA=0.067

**Figure 1.** Standardized structural model.

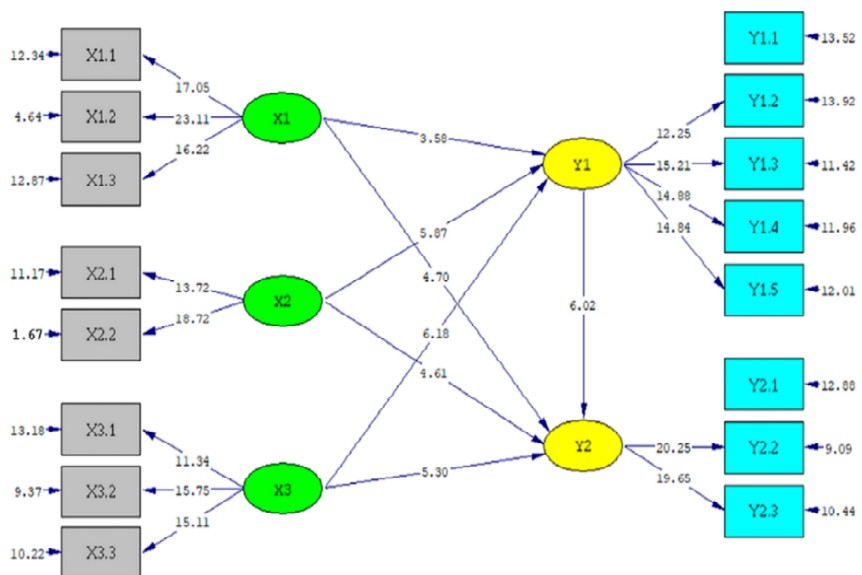

Chi-Square=288.04, df=94, P-value=0.00000, RMSEA=0.067

**Figure 2.** T-value structural model.

**Table 3.** Hypothesis testing results.

| Hypothesis | $\gamma/\beta$ | T-Value | Decision |
|---|---|---|---|
| $H_1$: Digital literacy ($X_1$) and professional performance ($Y_2$) | 0.22 ** | 4.70 | Supported |
| $H_2$: Grit ($X_2$) and professional performance ($Y_2$) | 0.21 ** | 4.61 | Supported |
| $H_3$: Instructional quality ($X_3$) and professional performance ($Y_2$) | 0.28 ** | 5.30 | Supported |
| $H_4$: Teaching creativity ($Y_1$) and professional performance ($Y_2$) | 0.35 ** | 6.02 | Supported |
| $H_5$: Digital literacy ($X_1$) and teaching creativity ($Y_1$) | 0.20 ** | 3.58 | Supported |
| $H_6$: Grit ($X_2$) and teaching creativity ($Y_1$) | 0.32 ** | 5.87 | Supported |
| $H_7$: Instructional quality ($X_3$) and teaching creativity ($Y_1$) | 0.37 ** | 6.18 | Supported |

** $p < 0.01$.

In addition, as presented in Table 4, this study also found a significant indirect (mediation) relationship between digital literacy, grit, and instructional quality and professional performance through teaching creativity ($\beta = 0.07$, $\beta = 0.11$, $\beta = 0.12$, $p < 0.01$). Instructional quality has a greater mediating role than others, while digital literacy has the smallest mediating role compared to the others. These results are consistent with the direct relationship between digital literacy, grit, and instructional quality and teaching creativity, which, respectively, from lowest to highest, are 0.20, 0.32, and 0.37. These results show that instructional quality and grit have a stronger direct and indirect relationship (mediation) than digital literacy.

**Table 4.** Mediation relationship analysis.

| | Mediation Relationship | β | Z-Value | Decision |
|---|---|---|---|---|
| 1. | Digital literacy ($X_1$) and professional performance ($Y_2$) through teaching creativity ($Y_1$) | 0.07 ** | 9.38 | Supported |
| 2. | Grit ($X_2$) and professional performance ($Y_2$) through teaching creativity ($Y_1$) | 0.11 ** | 9.23 | Supported |
| 3. | Instructional quality ($X_3$) and professional performance ($Y_2$) through teaching creativity ($Y_1$) | 0.12 ** | 9.40 | Supported |

** $p < 0.01$.

## 5. Discussion

The results of this research generally prove that digital literacy, grit, instructional quality, and teaching creativity are significantly related to teachers' professional performance, and digital literacy, grit, and instructional quality are also significantly related to teaching creativity. It can be shown in detail that the relationship between digital literacy and teachers' professional performance is positive. This indicates that when digital literacy, which includes digital competence, usage, and transformation, is carried out intensely, effectively, and efficiently, it can stimulate an increase in teachers' professional performance, especially in relation to subjects, didactics, and pedagogy. It means that digital literacy is a reasonable proclivity for teachers to have to improve their professional performance. These findings are consistent with and confirm the results of previous studies, which suggest that digital literacy has a significant relationship with teachers' professional performance [4,5,42–46]. Consequently, teachers need to increase their digital literacy to have greater opportunities to improve their professional performance.

This study also proves a positive relationship between grit and teachers' professional performance. This indicates that teachers with solid and stable grit, reflected in their consistency of interest and persistence in fighting for personal and organizational goals in the long term, tend to manage and develop their professional performance well. Thus, grit is an important determinant of teachers' professional performance. These findings are in line with and confirm the results of previous relevant research, which proves that grit makes a positive contribution to professional performance [6,7,22,60–64] and negates contradictory research results suggesting that grit has no significant effect on performance [16]. This has logical consequences for teachers, who must always manage and develop their grit thoughtfully and sustainably to improve their professional performance.

This study also reveals a positive relationship between instructional quality and teachers' professional performance. This empirical evidence shows that well-maintained instructional quality, especially classroom management, student support, and cognitive activation, can help improve teachers' professional performance, especially didactically and pedagogically. It indicates that teachers' instructional quality is an essential element in developing their professional performance. This finding is in line with previous studies that argue that instructional quality is closely related to teachers' professional performance [8,74]. Conse-

quently, teachers must strive to improve their instructional quality well and consistently to help improve their professional performance.

This study also proves the positive relationship between teaching creativity and teachers' professional performance. This empirical evidence confirms that when teaching creativity continues to be nurtured and developed, it will positively impact teachers' professional performance. Improving fluency, flexibility, originality, elaboration, and redefinition of various teaching tasks can help teachers master subject matter better and transfer it more efficiently and effectively. These findings are consistent and confirm research results that prove that teaching creativity influences teachers' professional performance [9,10,82–84] and provide insight for teachers to continually improve their teaching creativity as a modality to build their professional performance.

Teaching creativity, in this study, apart from being proven to influence teachers' professional performance, is also influenced by digital literacy, grit, and instructional quality. The influence is positive, thus indicating that all three are good predictors of teacher teaching creativity. This means that when teachers' digital literacy, grit, and instructional quality are improved or developed, they can increase teaching creativity. These findings confirm the results of previous research, which proves that digital literacy, grit, and instructional quality have a significant relationship with teachers' teaching creativity [13–15,89,90] and refute the results of some conflicting studies [17,18,21]. They can also inspire teachers to continuously improve their digital literacy capacity, grit, and instructional quality through various methods, approaches, and strategies, whether carried out independently on their own initiative or on the initiative and facilitation of the school, for example, through training activities or workshops. These findings also suggest that improving teachers' digital literacy capacity, grit, and instructional quality is not only the personal responsibility of teachers but also the responsibility of schools as beneficiary organizations for improving teachers' professional performance. Therefore, in this context, school principals and school authorities need to implement accelerated policies with strategic steps to help increase teachers' digital literacy capacity, grit, and instructional quality through concrete efforts, for example, facilitating training/workshop activities involving reputable experts.

This study also found new empirical evidence of the role of teaching creativity in mediating the relationship between digital literacy, grit, and instructional quality and teachers' professional performance. These findings are not only consistent and confirm the results of previous research, which were used as a reference to build this research hypothesis and negate the contradictory results of other research, but also promote a new empirical model of the causal relationship between digital literacy, grit, and instructional quality and teachers' professional performance with a mediating mechanism for teaching creativity. These findings, apart from providing theoretical contributions to studies in the field of technology and educational management, especially regarding teachers' professional performance seen from the perspective of digital literacy, grit, instructional quality, and teaching creativity, also provide practical implications for the practice of providing education in schools, especially in developing teachers' professional performance by utilizing the potential of digital literacy, grit, instructional quality, and teaching creativity. Therefore, the research findings deserve to be discussed critically, in depth, and comprehensively by practitioners, academics, and researchers before being adapted, modified, or adopted as pieces of material to support their work in the future. Utilizing the findings of this research should also not ignore its limitations, such as only using a single source (teachers), only using some of the indicators/theoretical dimensions found in the literature, only using a limited sample in a limited geographical location, and only using quantitative methods.

## 6. Conclusions

Schools need teacher professionalism to ensure the quality of educational output. This research found a significant relationship between digital literacy, grit, and instructional quality and teaching creativity and teachers' professional performance. Teaching creativity also has a significant relationship with teachers' professional performance and mediates the influence of digital literacy, grit, and instructional quality on teachers' professional performance. These findings promote a new empirical model of the causal relationship between digital literacy, grit, and instructional quality and teachers' professional performance through teaching creativity. They provide insight for teachers and school principals into collaborating to improve teachers' professional performance by developing digital literacy capacity, grit, instructional quality, and teaching creativity through specific mutually agreed-upon and realistically implemented activities, such as training and/or workshops. Meanwhile, researchers can conduct further research by considering additional data sources such as school principals and students, accommodating other indicators and dimensions not used in this research, taking samples from more expansive geographical locations, and adding qualitative or mixed (i.e., quantitative and qualitative) methods.

**Author Contributions:** Conceptualization, J.D. and W.W.; Methodology, W.W.; Software, W.W.; Validation, W.W.; Formal analysis, J.D. and W.W.; Investigation, J.D. and W.W.; Resources, J.D.; Data curation, W.W.; Writing—original draft, J.D.; Writing—review and editing, J.D.; Visualization, W.W.; Supervision, J.D. All authors have read and agreed to the published version of the manuscript.

**Funding:** This research received no external funding.

**Institutional Review Board Statement:** The study was conducted in accordance with the Declaration of Helsinki, and approved by the Academic Ethics Committee Board of the Postgraduate Faculty of lndraprasta University (code 1A, 16 February 2024).

**Informed Consent Statement:** Informed consent was obtained from all subjects involved in the study.

**Data Availability Statement:** The data presented in this study are available on request from the corresponding author.

**Conflicts of Interest:** The authors declare no conflicts of interests.

## Appendix A. Research Instrument

| Variables | Indicators | Items | CITCC | AC |
|---|---|---|---|---|
| Digital Literacy | DC | I learn various digital features independently when I have free time. | 0.635 | 0.897 |
| | | I have taken courses/training/workshops to improve my digital skills. | 0.441 | |
| | | I try to find the latest information on developments in digital technology. | 0.755 | |
| | DU | I utilize digital technology in the learning process. | 0.756 | |
| | | I apply various digital features easily. | 0.657 | |
| | | I use digital technology to make daily tasks easier. | 0.680 | |
| | DT | Digital technology makes me more creative. | 0.617 | |
| | | Digital technology encourages me to be more innovative in teaching. | 0.794 | |
| | | New features of digital technology stimulate me to be more adaptive to various educational challenges. | 0.683 | |

| Variables | Indicators | Items | CITCC | AC |
|---|---|---|---|---|
| Grit | COI | I am going to keep teaching until I retire. | 0.469 | 0.883 |
| | | Even though other careers offer more money, I intend to remain a teacher. | 0.660 | |
| | | Even though it will take time, I will persevere in pursuing my career to its peak. | 0.602 | |
| | | I will continue to be committed to completing my coursework. | 0.873 | |
| | POE | I make it a point to include teaching in my aspirations. | 0.764 | |
| | | I make long-term learning objectives a fighting orientation for one's life. | 0.752 | |
| | | My goal is to advance my career in a way that is sustainable. | 0.660 | |
| | | I will keep trying until I reach my academic objectives. | 0.755 | |
| Instructional Quality | CM | I use my study time allocation as efficiently as possible. | 0.416 | 0.901 |
| | | I try to prevent the possibility of chaos in class. | 0.489 | |
| | | I use clear rules during the learning process. | 0.780 | |
| | | I create a classroom atmosphere conducive to the learning process. | 0.582 | |
| | SS | In teaching, I pay attention to the specific characteristics of students. | 0.521 | |
| | | I give students the opportunity to differ in their opinions. | 0.676 | |
| | | I tolerate student mistakes during the learning process. | 0.668 | |
| | | I actively provide feedback on student assignments. | 0.532 | |
| | CA | I give students challenging assignments. | 0.491 | |
| | | I ask questions that invite student creativity. | 0.672 | |
| | | I actively explore students' reasoning capacity through critical questions. | 0.819 | |
| | | I stimulate students' critical thinking abilities through problem-solving-based learning. | 0.784 | |

| Variables | Indicators | Items | CITCC | AC |
|---|---|---|---|---|
| Teaching Creativity | Flue | When the learning model changes, I adjust quickly. | 0.738 | 0.859 |
| | | I can modify different educational materials to meet the needs of my students. | 0.503 | |
| | Flex | I let my students use as many different learning resources as they want. | 0.470 | |
| | | Though their thought processes differ from mine, I can still understand them. | 0.455 | |
| | Orig | I look for fresh approaches to teaching that meet real-world learning requirements. | 0.763 | |
| | | I always adapt my teaching strategies to the needs of my students. | 0.572 | |
| | Elab | Depending on the real circumstances of my students, I use a variety of teaching strategies. | 0.452 | |
| | | I select teaching resources that are suitable for the real-world learning environment. | 0.459 | |
| | Rede | To make sure the lesson material is still applicable to the situation today, I go over the whole thing. | 0.718 | |
| | | I reassess my past experiences with different learning strategies to make sure they still meet the demands of the learning process today. | 0.708 | |
| Professional Performance | Sub | I am an expert in the material I teach. | 0.555 | 0.912 |
| | | I regularly assess the subject matter. | 0.592 | |
| | | I frequently update the content. | 0.769 | |
| | Did | I employ a variety of teaching strategies. | 0.610 | |
| | | When presenting the material, I take into account the qualities of the students. | 0.819 | |
| | | When I teach, I consider the dynamics of the class. | 0.737 | |
| | Ped | When I teach, I consider how interested my students are in learning. | 0.903 | |
| | | Throughout the learning process, I consider the real state of the student's personality. | 0.750 | |
| | | I work with students to find solutions to a variety of learning challenges. | 0.583 | |

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
