# Peer review of "Unlocking Teacher Professional Performance: Exploring Teaching Creativity in Transmitting Digital Literacy, Grit, and Instructional Quality"

_education, doi:10.3390/educsci14040384_

Round 1
Reviewer 1 Report
Comments and Suggestions for Authors
Thank you for the opportunity to think through this paper! What you have presented here is a straightforward but effective study of the intersections between components including digital literacy, grit, and instructional quality and the larger concepts of professional performance and teaching creativity. The survey results you propose show that teachers find these concepts to be intertwined and important and I appreciate your discussion of this importance near the end of the paper. Overall, the piece is well written and easy to follow despite the complexities of the analysis you present. Well done!
That being said, there are a few issues related to the way you frame this study that must be addressed before publication. Largely, this boils down to more accurately defining and elaborating on the terms you focus on in the article. First and foremost, I’m not totally sure how you are conceiving of professional performance here. You say that “professional performance can be measured through three indicators, which are subject, didactic, and pedagogic” (p. 2, lines 68-70), but I don’t know what those indicators mean or how you conceive of them. Additionally, I do not understand the difference between instructional quality and professional performance. Would all three of those indicators you mention just define professional practice? If so, wouldn’t doing a study on the intersection between instructional quality and professional practice in teaching be somewhat redundant? I don’t think it is, but a clearer definition would help me better assess/understand the work.
Second, teaching creativity needs to be more thoroughly situated in the literature. There are lots of ways to define creativity and it seems like you are taking the definition of this term for granted. Essentially, it would be good to explain why you’ve decided to use your definition of creativity and not others (i.e. sociocultural or distributed definitions of the term). Moreover, there seems to be a conflation between teaching creativity and teaching for creativity. The latter term is used, but not well defined: what is meant by “scaffolding student’s emotional and creative responses”? (p. 4, lines 174-175) I would probably recommend just not bringing this term into the work, but it may still work if you want to clearly distinguish between these terms.
Third, the research questions need to be better defined. Specifically, I don’t know what the term “relates” means. It seems overly vague in a way that does not do justice to the analysis you have conducted! By defining this term (or, more accurately, using a term that directly relates to your research), you will more acutely orient us towards the research you have done.
Beyond these larger issues, a few smaller ones:
-The article could use another round of proofreading, especially in the first half. Nothing egregious, but there are a handful of small errors along the way that should be cleared up.
-You also bring up the phrase “teaching and learning process” on a few occasions. What do you mean by this? Lots of different ways to teach and learn! At the very least, I would clarify that you mean the kinds of teaching and learning that occur in schools (if this is what you mean).
-The research you present on grit largely ignores the ongoing critical research that has emerged around this term. Beyond merely questioning the impact or importance of grit, scholars have begun to see this concept as a harmful one that needs to be critically reimagined. This should be at least mentioned. The book “Debunking the Grit Narrative in Higher Education” and Slater’s piece in Critical Education are good places to start on this front. Crede’s piece in the Journal of Personality and Social Psychology may also be helpful.
-I’m confused why you listed the marital status of teachers in your description of the participants. Does this have some kind of effect on teaching? Seems odd.
Good luck with this project moving forward!
Comments on the Quality of English LanguageGenerally well written, but could use another pass in terms of proofreading. A few small errors scattered throughout.
Author Response
Dear Reviewers,
Many thanks for your willingness to make corrections and provide input for our article. It will help to improve the quality of our articles. We have tried hard to fulfill the reviewers' corrections, input, and expectations in the form of revisions, enrichments, additions, editing/proofreading of the manuscript, and explanations for certain parts. We hope that revising our article will meet the reviewers' expectations so that our article is worthy of publication in Education Sciences. As a note, the revised text in blue highlight is for Reviewer 1, and the green highlight is for Reviewer 2. Thank you very much for your attention, cooperation, and support.
Best regards,
Dr. Jafriansen Damanik
REVISION REPORT: REVIEWER 1
(Revision in Blue Hightlight)
|
Subject |
Correction/Comments |
Revision/Explanation |
|
Definition of instructional quality and professional performance |
I’m not totally sure how you are conceiving of professional performance here. You say that “professional performance can be measured through three indicators, which are subject, didactic, and pedagogic” (p. 2, lines 68-70), but I don’t know what those indicators mean or how you conceive of them. Additionally, I do not understand the difference between instructional quality and professional performance. Would all three of those indicators you mention just define professional practice? If so, wouldn’t doing a study on the intersection between instructional quality and professional practice in teaching be somewhat redundant? I don’t think it is, but a clearer definition would help me better assess/understand the work. |
The definition of instructional quality and professional performance, including indicators of professional performance, has been improved. |
|
Definition of teaching creativity |
Teaching creativity needs to be more thoroughly situated in the literature. There are lots of ways to define creativity and it seems like you are taking the definition of this term for granted. Essentially, it would be good to explain why you’ve decided to use your definition of creativity and not others (i.e. sociocultural or distributed definitions of the term). Moreover, there seems to be a conflation between teaching creativity and teaching for creativity. The latter term is used, but not well defined: what is meant by “scaffolding student’s emotional and creative responses”? (p. 4, lines 174-175) I would probably recommend just not bringing this term into the work, but it may still work if you want to clearly distinguish between these terms. |
It has been improved. |
|
Hypothesis |
The research questions/hypothesis need to be better defined. Specifically, I don’t know what the term “relates” means. It seems overly vague in a way that does not do justice to the analysis you have conducted! By defining this term (or, more accurately, using a term that directly relates to your research), you will more acutely orient us towards the research you have done. |
It has been improved. |
|
Proofreading |
The article could use another round of proofreading, especially in the first half. Nothing egregious, but there are a handful of small errors along the way that should be cleared up. |
Manuscript has been revised and proofread by the Professional Editing Service from MDPI. |
|
Phrase teaching and learning |
You also bring up the phrase “teaching and learning process” on a few occasions. What do you mean by this? Lots of different ways to teach and learn! At the very least, I would clarify that you mean the kinds of teaching and learning that occur in schools (if this is what you mean). |
It has been improved. |
|
Critical issue of grit |
The research you present on grit largely ignores the ongoing critical research that has emerged around this term. Beyond merely questioning the impact or importance of grit, scholars have begun to see this concept as a harmful one that needs to be critically reimagined. This should be at least mentioned. The book “Debunking the Grit Narrative in Higher Education” and Slater’s piece in Critical Education are good places to start on this front. Crede’s piece in the Journal of Personality and Social Psychology may also be helpful. |
It has been added. |
|
Marital status |
I’m confused why you listed the marital status of teachers in your description of the participants. Does this have some kind of effect on teaching? Seems odd. |
Marital status has been deleted from the manuscript. |
REVISION REPORT: REVIEWER 2
(Revision in Yellow Hightlight)
|
Subject |
Correction/Comments |
Revision/Explanation |
|
Introduction
|
There are many references to the latest research referenced in this manuscript. However, I still need to see your theoretical framework. What research gaps did you find that made this research enjoyable? It would help if you wrote it in the introduction. I didn't see that. |
The theoretical framework has been added. Research gaps found in a number of sources have been cited in the Introduction section. |
|
Method
|
When I read the title Teacher Professional Performance, what came to my mind was research where the researcher made direct observations while the teacher was teaching. I expected your research process to involve something other than distributing questionnaires and conducting analysis in the research room. How do you know the truth of Master's answer? How do you describe "I utilize digital technology in the learning process"? How skilled is the teacher in using technological devices? These questions need to be explained.
|
"From the start, this research was designed using quantitative methods where data was obtained using a questionnaire in self-report format with a single data source (teacher). Under these conditions, qualitative information was not revealed in this research. However, to ensure the degree of truth (unbiasedness) of the data obtained from teachers, a common method bias (CMB) test was carried out to detect the possibility of data bias in this research. The results show that there is no data bias (CMB) phenomenon. Under these conditions, the quantitative research data obtained from teachers through questionnaires (self-report) does not contain data bias, so there is no need to doubt and be worthy of trust." |
|
Discussion
|
In your discussion, you explain the analysis of statistical results well, but what the process of providing instruction to students is. Student responses, the teacher's reflections after teaching, and whether the instruction was right or wrong. I didn't find anything in your discussion. The limitations of your research should be explained before conclusion. |
This research was designed using quantitative methods, using a questionnaire instrument (self-report) with a single data source (teacher) so that it does not reveal and discuss qualitative facts related to all research variables and the relationships between them. This is an added limitation of this research. |
|
Tthe Quality of English Language
|
Minor editing of English language required.
|
Manuscript has been revised and proofread by the Professional Editing Service from MDPI. |
|
Qualitative research |
It should be qualitative research than quantitative. There so many question that can be answered by qualitative research in this field. |
The qualitative research proposal is interesting and worthy of consideration for future research. This has been added to the end of the conclusion as a suggestion. |

Reviewer 2 Report
Comments and Suggestions for Authors
Introduction
There are many references to the latest research referenced in this manuscript. However, I still need to see your theoretical framework. What research gaps did you find that made this research enjoyable? It would help if you wrote it in the introduction. I didn't see that.
Method
When I read the title Teacher Professional Performance, what came to my mind was research where the researcher made direct observations while the teacher was teaching. I expected your research process to involve something other than distributing questionnaires and conducting analysis in the research room. How do you know the truth of Master's answer? How do you describe "I utilize digital technology in the learning process"? How skilled is the teacher in using technological devices? These questions need to be explained.
Discussion
In your discussion, you explain the analysis of statistical results well, but what the process of providing instruction to students is. Student responses, the teacher's reflections after teaching, and whether the instruction was right or wrong. I didn't find anything in your discussion. The limitations of your research should be explained before conclusion.
Comments on the Quality of English Language
Minor editing of English language required.
But, it should be qualitative research than quantitative. There so many question that can be answered by qualitative research in this field.
Author Response

(The authors gave the same response as above.)

Round 2
Reviewer 1 Report
Comments and Suggestions for Authors
Overall, you have done a wonderful job in responding to the comments that the other reviewer and myself have brought up. I appreciate the thoroughness and the detail with which you have done so. The paper and its contributions to the field are now much clearer.
While I am recommending this piece for publication, there are a few smaller revisions that should be made before doing so:
· I would move definition of professional performance to the very beginning of the introduction where it is first mentioned. Readers who are unfamiliar with the term, such as myself, won’t know what you are talking about until the literature review section if you don’t.
· Are your definitions of subjects, didactics, and pedagogy from the source you cite? If so, I would be fine leaving them. But that is a strange definition of pedagogy, as most of the ones I am familiar with are more similar to how you define didactics. Maybe subjects, pedagogy, and context?
· Glad you included the reference to critical literature related to grit, but I would justify it further. If grit is a racist theory, why still use it? Again, there are reasons to explore and include grit here, but there needs to be some justification as to why in the writing.
· Section 4.1 feels like it should be in the methods section, since its describing the analysis you conducted and not the results. Not a necessary change, but one worth considering.
· I’m not sure how you are defining a theoretical framework in this paper. I appreciate you trying to respond to the other reviewer, but how it is presented is a bit muddled. Maybe something like “the intersection between professional performance and teaching creativity”? Then you describe all of the other ideas that serve as mediating influences?
Well done, and congratulations on writing a wonderful paper!
Author Response
Dear Reviewer,
Thank you very much for your appreciation of the revision of our article and your willingness to provide corrections and input, which are critical and valuable for improving the quality of our article. As with the previous revision, we have enthusiastically and carefully carried out improvements this time to meet the expectations of reviewers and editors and the standards of the Education Sciences journal. As a note, the revised text in green highlight is for Reviewer 1. Thank you very much for your attention, cooperation, and support.
Best regards,
Dr. Jafriansen Damanik
REVISION REPORT: REVIEWER 1 R2
(Revision in Green Highlight)
|
Subject |
Correction/Comments |
Revision/Explanation |
|
Definition of professional performance in beginning of the introduction |
I would move the definition of professional performance to the very beginning of the introduction, where it is first mentioned. Readers who are unfamiliar with the term, such as myself, won’t know what you are talking about until the literature review section if you don’t. |
We have added a definition of professional performance at the beginning of the introduction. |
|
Definition of didactis and pedagogy |
Are your definitions of subjects, didactics, and pedagogy from the source you cite? If so, I would be fine leaving them. But that is a strange definition of pedagogy, as most of the ones I am familiar with are more similar to how you define didactics. Maybe subjects, pedagogy, and context? |
Definitions of subjects, didactics, and pedagogy according to the quoted source. |
|
The reasons to explore of grit |
Glad you included the reference to critical literature related to grit, but I would justify it further. If grit is a racist theory, why still use it? Again, there are reasons to explore and include grit here, but there needs to be some justification as to why in the writing. |
We have added the rationale for using the grit variable in the article. |
|
Move Section 4.1 in the methods section |
Section 4.1 feels like it should be in the methods section, since its describing the analysis you conducted and not the results. Not a necessary change, but one worth considering. |
Section 4.1 has been moved to the methods section. |
|
Defining a theoretical framework |
I’m not sure how you are defining a theoretical framework in this paper. I appreciate you trying to respond to the other reviewer, but how it is presented is a bit muddled. Maybe something like “the intersection between professional performance and teaching creativity”? Then you describe all of the other ideas that serve as mediating influences? |
It has been improved. |
